# EEG-Based Emotion Classification Using Stacking Ensemble Approach

**DOI:** 10.3390/s22218550

**Published:** 2022-11-06

**Authors:** Subhajit Chatterjee, Yung-Cheol Byun

**Affiliations:** 1Department of Computer Engineering, Jeju National University, Jeju 63243, Korea; 2Department of Computer Engineering, Major of Electronic Engineering, Institute of Information Science & Technology, Jeju National University, Jeju 63243, Korea

**Keywords:** deep learning, emotion classification, EEG data, stacking ensemble classifier, random forest, lightGBM, gradient boosting classifier

## Abstract

Rapid advancements in the medical field have drawn much attention to automatic emotion classification from EEG data. People’s emotional states are crucial factors in how they behave and interact physiologically. The diagnosis of patients’ mental disorders is one potential medical use. When feeling well, people work and communicate more effectively. Negative emotions can be detrimental to both physical and mental health. Many earlier studies that investigated the use of the electroencephalogram (EEG) for emotion classification have focused on collecting data from the whole brain because of the rapidly developing science of machine learning. However, researchers cannot understand how various emotional states and EEG traits are related. This work seeks to classify EEG signals’ positive, negative, and neutral emotional states by using a stacking-ensemble-based classification model that boosts accuracy to increase the efficacy of emotion classification using EEG. The selected features are used to train a model that was created using a random forest, light gradient boosting machine, and gradient-boosting-based stacking ensemble classifier (RLGB-SE), where the base classifiers random forest (RF), light gradient boosting machine (LightGBM), and gradient boosting classifier (GBC) were used at level 0. The meta classifier (RF) at level 1 is trained using the results from each base classifier to acquire the final predictions. The suggested ensemble model achieves a greater classification accuracy of 99.55%. Additionally, while comparing performance indices, the suggested technique outperforms as compared with the base classifiers. Comparing the proposed stacking strategy to state-of-the-art techniques, it can be seen that the performance for emotion categorization is promising.

## 1. Introduction

A physiological and behavioral response to internal and external stimuli, emotion is a complicated, physiological behavior of human beings [1]. The goal of human emotion recognition is to identify human emotions from a variety of modalities, including body language, physiological signs, and audiovisual manifestations. In human-to-human communication and contact, emotion is crucial. Emotion is a result of mental processes that people engage in, and it can be expressed as a reflection of their psychophysiological states [2].

Over the past few years, numerous investigations on engineering strategies for automatic emotion identification have been conducted. Three broad categories can be made for them. The first category examines speech, body language, and facial expressions [3,4,5,6]. These audio–visual methods enable non-contact emotion detection. The second group largely focuses on physiological signals at the periphery. Studies have shown that different emotional states alter peripheral physiological signals. The third group of methods focuses primarily on brain signals obtained from the central nervous system using a device that records the brain signals, known as an electroencephalography (EEG) or electrocorticography (ECoG). It has been demonstrated that EEG signals have informative characteristics in response to emotional states [7,8] among these brain signals. According to Davidson et al. [9], the experience of two emotions was correlated with frontal brain electrical activity; they were positive and negative. According to the studies, there has been a lot of discussion regarding the link between EEG asymmetry and emotions.

ECG signal offers useful information for identifying emotional stress in people. For many years, research has been conducted on emotional stress, mostly in the psychological realm. Emotional stress is a major contributor to mental illnesses such as depression, anxiety, and bipolar disorders. Positive emotions (such as happiness and surprise) and negative emotions are two major categories for emotions (sad, anger, fear, and disgust). EEG provides a good temporal resolution when measuring the brain’s electrical activity. In this study, we classify three states of human emotion; they are positive, negative, and neutral. Understanding that brain activity is subject-specific and that people’s emotional brain activities in various brain areas vary for a given subject is crucial to understanding how to recognize emotions through brain activity. This article discusses a better way of categorizing human emotions from EEG data.

Determining how fleeting mental experiences are translated into a specific pattern of brain activity is a significant difficulty in brain–machine interface applications. The quantity of information required to accurately represent the many states of the EEG signals, which are complex, non-linear, and unpredictable, is one of the critical problems with classifying EEG signals. This paper proposes a new ensemble model developed with a random forest, light gradient boosting machine, and a gradient boosting-based stacking ensemble (RLGB-SE) model to categorize various emotional states. The 2549 source attributes created a smaller dataset through feature selection. The methods of choice scored the qualities according to how well they performed in classification, and a manual cutoff point was tuned when the score started to decline, keeping only the strongest traits. A stacking ensemble classifier combines different classification models to improve the model’s accuracy. To build the model, we employed three classifiers as foundation models: random forest, light gradient boosting machine, and gradient boosting classifier. These three classifiers’ outputs serve as the meta-input. The paradigm separates mental states into three categories: positive, negative, and neutral. The proposed RLGB-SE performs better than cutting-edge techniques for identifying intelligent emotions in EEG signals.

The key contributions of this work are outlined in the following points:An advance ensemble model developed with random forest, light gradient boosting machine, and gradient-boosting-based stacking ensemble (RLGB-SE) classifier models is proposed to classify various emotional states.A new method of categorizing emotions into three categories (positive, neutral, and negative) allows for real-world mental states that are not primarily characterized by emotions.We conduct a comparison to validate the performance of the proposed ensemble model with advanced ML models and various ensemble model combinations.

The layout of the article is organized into the following sub-sections. Section 2 explains earlier related experiments that used EEG data to classify human emotions in detail. The methodology is described briefly, and the model description is illustrated in Section 3. Section 4 thoroughly explains all the findings in EEG signal classification and includes a comparison with current machine learning models. Finally, Section 5 brings this paper to a close.

## 2. Related Works

The recognition of human emotions via EEG signal data has been the subject of extensive research in recent years. In earlier investigations, various feature extraction techniques, channel selection strategies, and classification techniques have been used to identify emotions. Machine learning methods and statistical variables acquired from EEG data are frequently combined with categorizing mental states [10,11]. For finite control points, these mental states can serve as a brain–computer interface. A Muse headband gained the respect of neuroscientists for its efficiency, affordability, and classification accuracy.

The article by Alhagry et al. [12] classified the DEAP dataset into high/low arousal, high/low valence, and high/low liking classes with average accuracies of 85.65%, 85.45%, and 87.99% using the long short-term memory neural network and all of the EEG data. Using multi-channel EEG-based emotion recognition with deep forest, Cheng et al. [13] classified emotions using the EEG data from the DEAP and DREAMER datasets. They utilized deep forest to classify the data of all the channels into high/low valence, high/low arousal, and high/low dominance. They then mapped the data of all the channels to 2D frame sequences. Similar to this study, another study [14] classified emotions using the whole DEAP dataset and a transfer learning strategy. EEG-based emotion identification models for the three emotions of positive, neutral, and negative were built using deep belief networks (DBNs). It achieved the best outcome when compared to SVM, LR, and KNN machine learning models, with an accuracy of 86.5% [15]. Another method for EEG-based emotion recognition focuses on how subjects’ EEGs behave differently while watching films intended to evoke positive or negative emotions [16]. SVM and KNN models produced the best classification accuracy. The accuracy and effectiveness of EEG-based emotion classifiers are also improved by the feature smoothing approach known as the linear dynamical system (LDS), and the feature selection algorithm known as the minimal-redundancy-maximum-relevance (MRMR) algorithm. To take the first step toward a potential EEG-based brain–computer interface (BCI) for supporting autism intervention, Fan et al. [17] investigated the viability of detecting the engagement level, emotional states, and mental effort during VR-based driving using EEG. The Bayes network, naive Bayes, support vector machine (SVM), multilayer perceptron, K-nearest neighbors (KNN), random forest, and J48 classification techniques were used and compared. The classification results were encouraging, with over 80% accuracy in assessing engagement and mental workload and over 75% in classifying emotional states.

In addition to using EEG asymmetry to examine emotion, researchers also looked at the relationship between EEG and emotion using event-related potentials, which index a small percentage of mean EEG activity [18,19,20]. These methods still have two drawbacks, though. The first is that EEG characteristics must be averaged in the current methods. As a result, they require more significant periods to identify the emotional state from EEG signals. The ability to record only a tiny amount of EEG activity is the other. Due to these drawbacks, the current approaches to evaluating emotional states are either inappropriate or insufficient for use in practical contexts.

Since EEG reflects different types of cognitive activity in the brain and emotional states, given the variability of EEG and electrode locations, it is not always clear which independent variables to utilize to differentiate between moods. Thus, in recent years, scientists have attempted to employ more sophisticated techniques to discover the relationship between emotional shifts and EEG data. Chanel et al. [21] suggested an emotion detection system that classifies two emotional states using EEG. Their study had a 72% naïve Bayes classification accuracy for the arousal component of emotions and a 70% Fisher discriminant analysis classification accuracy. Li et al. [22] classified the feelings of happiness and sadness using EEG data. They used linear-SVM and common spatial patterns (CSP) in the experimental setup. Their analysis found that two emotional states had a favorable recognition rate of 93.5%. Using EEG features, Zhang et al. [23] classified the subject’s status into two emotional states, positive and negative, with an average accuracy of 73.0%. The study examined and categorized emotional states elicited by a natural setting using electroencephalography (EEG) and functional magnetic resonance imaging (fMRI). The Laplacian filtering method was utilized to pre-process the raw EEG data, and KNN and linear discriminant analysis (LDA) methods were used to classify the emotional states. Discrete wavelet transform was then used to split the raw EEG signals into three distinct frequency bands. An emotion recognition method using several EEG channels was demonstrated by Murugappan et al. [24], who achieved 83.26% of accuracy for five emotional states. A system for user-independent emotion identification was presented by Petrantonakis et al. [25]. An SVM classifier had an 83.33% recognition rate for six different emotion categories. The suggested classifier achieved 87% accuracy using the Muse headband EEG dataset with a combination of cross validation and several feature selection techniques [26]. In later work proposed by Jordan et al. [27], three states of emotion classification were studied using an ensemble approach, and the proposed model obtained 97.89% accuracy. An ensemble strategy was presented [28] to classify Parkinson’s disease (PD), including feature selection and a sample dependent classifier. The brain works seem to change from person to person and from one emotional state to another. Using EEG data associated with PTSD, a hybrid deep learning model combining CNN-LSTM and ResNet-152 models was created to categorize emotion [29]. Classification model prediction performance was improved via ensemble learning. A semisupervised multiple choice learning (SemiMCL) strategy was used in the study to improve the assignment of labeled data among the constituent networks and to take advantage of unlabeled data to gather domain-specific information [30].

## 3. Methodology

Brain science has grown in importance as technology and science have advanced. Affective science has long debated how to categorize emotions or how to compare and contrast one feeling with another. The emotion classification of EEG data has received a lot of attention due to the rapid development of various machine learning algorithms, each of which has its advantages and disadvantages. A stacking ensemble classifier is present here that combines different classification models to improve the model’s accuracy. We used the training data to train parallel classification models. It is preferable to use a variety of models to provide the most significant results. Since everything is assumed to be independent of one another, we trained models using various classification techniques, including RF, lightGBM, and GBC. We propose a new ensemble classifier using a random forest, light gradient boosting machine, and gradient boosting-based stacking ensemble (RLGB-SE) classifier. The experimental procedure is elaborated in Figure 1. The batch data were preprocessed, and feature engineering methods were employed. The proposed model was then trained using a set of features. Finally, we classified the emotions using a variety of evaluation parameters.

### 3.1. Data Analysis

We used the Muse headband EEG brainwave data, which have three kinds of emotional states: positive, negative, and neutral. Our primary focus is to classify these three emotional stages correctly. Our goal is to identify discriminative EEG-based features and suitable classification techniques that can classify brainwave patterns according to the degree of activity that they exhibit. The measuring of electrical activity generated by the brain is known as electroencephalography (EEG) [31]. The electrodes used to read the electrophysiological currents produced by the brain owing to nerve activity are used to acquire EEG data. Microvolts (uV), which are used to measure raw electrical data, over time produce wave patterns.

The data we used in this experiment are available online in Kaggle since the dataset of EEG brainwave data were processed according to Jordan et al. [27,32]. For collecting the data, a Muse EEG headband with four electrodes corresponding to the international EEG placement standard’s TP9, AF7, AF8, and TP10 reference sites was used to collect data for the experiment. The 10–20 international system [33] of electrode placement is a technique for describing where scalp electrodes should be placed. The 10–20 standard’s original design used just 21 electrodes; a typical interelectrode distance is 6 cm. T stands for temporal, P stands for parietal, F stands for frontal, and A stands for auricular or ear electrodes. These depict electrode placement in the skull. Even-numbered electrodes (8,10) refer to electrode placement on the right side of the brain, and odd numbers (7,9) refer to those on the left side of the brain. Experimental data were gathered from two individuals—a man and a female aged between 20 and 22 years—for three minutes in each of the three states of positive, neutral, and negative, which means sixty seconds of data. The emotional information from six movie clips, which produced 12 min (720 s) of each emotional brain activity data, is detailed in Table 1.

The primary problems in brain–computer interface applications are EEG signal feature extraction and categorization. The complexity of the data is a complicated issue for EEG feature extraction. The best course of action is to use strategies that satisfy all the requirements because the signals are only thought to be stationary for brief periods. Feature selection as per this dataset contains EEG brainwave data that have been extracted using established statistical feature-extraction techniques [27,32,34].

Feature selection must be carried out to find valuable statistics and simplify the model development procedure. This will save time and computational resources for the training and classification operations. The goal of feature selection is to eliminate useless information that only helps raise the demand for resources. Usually, generated EEG data are vast. We must select the useful values by reducing them to smaller data using the feature selection method. We have 2132 number of rows and 2548 number of columns for our experiment, where the label column denotes three types of emotions. Figure 2 depicts three emotions of the EEG signal.

### 3.2. Design of Stacking Ensemble Model

Creating a combination of models with ensemble machine learning entails training individual models on resampled datasets [35]. After that, these models are integrated to produce forecasts for the desired variables. Ensemble methods are widely employed in classification and regression problems [36] to achieve good performance. Ensemble machine learning’s core tenet is that by merging many models, the mistakes of one base learner will probably be made up for by other base learners. Because of this, the ensemble’s total prediction performance will be superior to that of a single base learner [37]. Classifiers whose decisions are combined to form the ensemble from the classification viewpoint are called base classifiers [38].

Our ensemble model was developed using the open-source Python module “pycaret”. A supervised machine learning module called pycaret classification (pycaret. classification) categorizes elements into binary category using various methods and algorithms. Binary or multi-class classification issues can be solved with the pycaret classification module. The pycaret classification module mainly covers sophisticated techniques, such as stacking, ensembling, and hyperparameter tuning. The training accuracy was gathered from different modules. The top three models were then chosen, and they were used for stacking. The hyperparameter parameters for the base models, random forest, lightGBM, and GBC, are listed in Table 1. As a meta-model, the random forest classifier is employed.

Stacking ensemble classifier [39] algorithms are a technique for improving the classification accuracy. In this work, the stacking ensemble model employs three base classifiers, including RF, LightGBM, and GBC, at base level 0. The RF was employed as a meta classifier at meta level 1. We initially created the base model by instructing three base classifiers on the entire training input set of data. Each base model’s prediction serves as the input for the meta-model classifier [40]. Next, a new feature representation for the meta level classifier was created using the output predictions of base classifiers at level 1. As seen in Figure 3, the use of meta classification during the final judgment step allows for the final prediction of class labels. We fed the training data (TD) into the suggested stacking ensemble (RLGB-SE) classifier model in order to categorize different emotional states. A k-fold cross-validation technique is frequently employed while training the classifiers to prevent the issue of over-fitting. The k-fold cross-validation approach is crucial in determining how well classifiers predict the future. The training data (TD) are split into a set of k disjoint subsets of equal size ($TD_{1}$, $TD_{2}$, $TD_{3}$, …,$TD_{K}$) for k-fold cross validation. In this study, three base classifiers, such as RL, lightGBM, and GBC, are trained at the base level (level 0), while 10-fold cross validation is considered. As a meta-model, the random forest classifier is used (level 1). Algorithm 1 lists the phases that make up the classification process for the suggested stacking ensemble model.

The complexity of an algorithm is referred to as computational complexity. It explicitly focuses on time and memory needs, and measures the quantity of resources needed to run the algorithm. The difficulty of the most effective algorithms that can be used to tackle a problem is simply its computational complexity. First, we create a stacking ensemble classification algorithm to incorporate low-complexity, high-diversity base classifiers. The meta-classifier chooses the fundamental parameter to lessen the complexity of the algorithm. The performance of EEG-based emotion classification is enhanced using a technique. Increased processing time is one of the side effects of increased complexity. A too-complicated model will take a long to run when the amount of data available increases exponentially. For the stacking model to optimize and perform better, there must be a greater variety of models in the ensemble. We used the random forest as a meta-classifier and have the following approximations for the computational complexity shown in Equation (Equation 1), where n = number of training examples (rows), d = number of data features (columns), k = number of neighbors:(1)$$Training_{TimeComplexity}=O\left(n^{*}\log{\left(n\right)}^{*}\;d^{*}k\right)$$
**Algorithm 1 **Pseudocode of the proposed stacking ensemble strategy.1:**Input: Load actual Training data (TD)**2:**Output: Prediction from the ensemble classifier (EC)**3:Step 1: When creating the training set for classifiers, use cross-validation (k-fold).4:Step 2: Divide dataset (TD) into k equal-sized subsets at random: TD = ($TD_{1}$, $TD_{2}$, $TD_{3}$, …$TD_{K}$); where (k = 10)5:Step 3: For k <− 1 to K6:(level 0) Trained base classifiers namely RF, LightGBM and GBC;7:For n <- 1 to N8:Learn classifiers from TD or $TD_{K}$9:End for10:Step 4: Create a training set for the RF meta classifier;11:Step 5: (level 1) Train meta classifier RF;12:From the created training set, learn a new classifier.End for13:Step 6: From ensemble classifier, return EC;

#### 3.2.1. Random Forest

Random forest (RF), developed by Breiman et al. [41] in 2001, is a supervised machine learning algorithm in which decision trees are constructed during the training of the dataset to produce the final result. It is a well-liked tree-based ensemble machine learning method that can be applied to both big and small issues for classification and regression tasks. RF is a classification technique that uses ensemble learning, consisting of several decision trees (DT). Each DT offers a categorization for the incoming data to categorize a new instance. RF then chooses the most popular guess after compiling the categories. Each tree receives sampled data from the original dataset as input. Additionally, a subset of features from the optional characteristics is randomly chosen to develop the tree at each node. RF is a combination of tree predictors, where each tree is reliant on values of a random vector collected independently, and all trees in the forest have the same distribution with n-sized tuples taken from the training data set with replacement, followed by an arrangement of the trees. To produce k random trees, repeat this step k times. To construct the DTs in the forest, RF utilizes bagging. The trees were trained using roughly two-thirds of the training data, and the remaining one-third was utilized for internal cross validation to estimate the model’s performance [42,43]. The class assignment probabilities obtained by all created trees were averaged to arrive at the final classification result. Our RF analysis was carried out in this work using the pycaret Python library, using the following final parameters shown in Table 2.

#### 3.2.2. Light Gradient Boosting Machine

Ke et al. [44] proposed different approaches of the gradient boosting algorithm that grows trees vertically by using a leaf-wise algorithm. The main benefit of lightGBM is that it can train more quickly and effectively than many other algorithms. It streamlines the training process by discretizing continuous feature data into buckets. By using discrete bins instead of continuous data, memory use is decreased. The key to achieving higher accuracy is to use a leaf-wise split strategy rather than a level-wise split approach, which results in far more complicated trees. Large dataset compatibility is improved. Large datasets can be used with the same success, greatly lowering training time. Our LightGBM analysis was carried out in this work using the pycaret Python library, using the following final parameters shown in Table 2.

#### 3.2.3. Gradient Boosting Classifier

Friedman et al. [45] proposed gradient boosting techniques which have been shown to outperform other approaches on complex and inevitably unbalanced classification tasks, often requiring much less labeled data and requiring much less training effort. Gradient boosting is based on the iterative fitting of the residuals derived from the successive training and forecasts produced by weak learner algorithms. The final prediction for the class with the highest probability comes from the ensemble of decision trees convergent on the smallness of residuals or when the maximum number of trees is reached. Using the pycaret Python library, we calculated its results using the following final parameters shown in Table 2.

After splitting the data into two parts, the data must be accurately labeled for the classifier to be adequately trained. The staking ensemble classifier was trained using one part of the data and tested with the other. To assess the suggested model’s accuracy, we used a variety of metrics to evaluate the performance, including accuracy, recall, precision, and F1 score. We investigated the model further using the confusion matrix, receiver operation characteristics (ROC), and other measures. TPR (true positive rate) and FPR (false positive rate) are correlated, as seen by the ROC curve. Treating each label indicator matrix component as a potential binary prediction, a process known as micro-averaging, ROC, and AUC can be used for multi-label classification as a standard binary classification metric.

## 4. Results

The following subsections provide and explain the specifics of the findings from the experiments carried out for this study.

### 4.1. Comparison with Confusion Matrix

In the confusion matrix, the primary diagonal element stands for the accurate classification of testing samples, whereas components in the off-diagonal signify inaccurate classification. The x-axis represents the actual emotions, and the y-axis represents the predicted emotions. The results of the confusion matrix show that, between the two classifiers, the voting classifier misclassified seven cases. For the suggested strategy, just three samples were misclassified. In light of the findings mentioned above, it can be said that the suggested stacking ensemble classifier has a high overall classification accuracy (99.55%) compared to the voting classifier’s accuracy level (98.91%). The proposed stacking classifier model and the voting classifier’s confusion matrix are shown in Figure 4.

Using the ROC (receiver operating characteristics) curve to demonstrate the efficacy of the suggested model for multiclass classification is preferable [46]. A ROC plot is used to assess a model’s performance between sensitivity and specificity. Sensitivity is the ability to recognize entries that fall within the positive class correctly. Accurately recognizing entries that belong in the negative class is what is meant by specificity. They are depicted in separate colors in Figure 5. The proposed model, it should be noted, performs better on both the micro and macro ROC curves.

### 4.2. Classification Result Compared with Different Models

The classification accuracy denoted by $C_{Accuracy}$ of classifiers expressed by Equation (Equation 2). It can be defined as the number of emotions correctly detected among the total number of emotions studied. The total number of instances is 2132. For training, we used 70% of the data, that is, 1492 instances, and 30% of the data, that is, 640 instances, for testing purposes.

It can be stated as the percentage of accurately predicted positive observations among all of the observations in the class is defined in Equation (Equation 3). The following information in Equation (Equation 4) describes the percentage of successfully predicted positive observations among all predicted positive observations. The F1-measure, which can be characterized as a weighted average of recall and precision, is defined in Equation (Equation 5).

A $T_{p}$ is a result where the model properly predicts the positive class, where $T_{p}$ stands for true positive, $T_{n}$ for true negative, $F_{p}$ for false positive, and $F_{n}$ for false negative. Tn represents a result where the model accurately predicted the negative class. $F_{p}$ is a result in which the model incorrectly predicted the positive class. $F_{n}$ is a result in which the model incorrectly predicted the negative class. The values of the aforementioned evaluation measures determined for a different model are displayed in Table 3.
(2)$$C_{Accuracy}=\frac{\left(T_{p}+T_{n}\right)}{\left(T_{p}+T_{n}+F_{p}+F_{n}\right)}$$
(3)$$Recall=\frac{T_{p}}{\left(T_{p}+F_{n}\right)}$$
(4)$$Precision=\frac{T_{n}}{\left(T_{n}+F_{p}\right)}$$
(5)$$F-Measure=2\cdot \frac{P\cdot R}{\left(P+R\right)}$$

Our proposed model was contrasted with various cutting-edge machine learning classifiers. Our comparisons include GBC, LightGBM, RF, extra trees, decision trees, ada boost, K neighbors, and voting in addition to the suggested stacking ensemble classifier. According to Table 3 classification accuracy results, the proposed stacking ensemble classifier outperforms the base classifiers GBC (98.73%), LightGBM (98.73%), and RF (98.59%) with a classification accuracy score of 99.55%. Even though some of these models work better when used alone, a stacking ensemble learns the most effective approach to integrate the predictions from various efficient machine learning models. With respect to the suggested model, we contrasted the performance of several models and voting classifiers. Table 3 compares the F-measure, recall, accuracy, and precision. GBC, LightGBM, and RF coped the best when comparing the result with other models. So, for building the base of the stacking ensemble model, we used these three classifiers and used RF as a meta-classifier in this approach. We discovered that merging several of the most effective ML models can produce even better results. When evaluated using several evaluation metrics, the suggested model outperforms the individual models.

This study used EEG data to categorize human emotions into three categories: positive, negative, and neutral. The classification accuracy results for various machine learning models are displayed in Table 4. According to much research, the whole EEG dataset can accurately classify emotions. The overall findings demonstrate that, when applied to EEG data, the proposed model predicts with a greater classification accuracy than other models.

## 5. Conclusions

In this article, we provide a stacking ensemble method for classifying emotions. We use three different emotions for classification tasks to illustrate the value of our suggested approach. The literature then presents compelling evidence that the ensemble technique outperforms other models. First, the classifiers (RF, LGBM, and GBC) were trained at the proposed ensemble model’s base level (level 0). Second, we initially created the base model by instructing three base classifiers on the entire training input data set. Each base model’s prediction serves as the input for the meta-model classifier (RF). Next, a new feature representation for the meta-level classifier was created using the output predictions of base classifiers at level 1. The performance was evaluated by the accuracy, precision, recall, F-measure, and ROC area of the ensemble classifier. According to the classification accuracy result, the proposed ensemble model achieves a superior classification accuracy of 99.55%.

Our usage of only four channels and the small amount of data we obtained is the work’s limitations. We want to test our model with a bigger dataset in the future. This study’s biggest flaw is the small number of patients included in the dataset. Future work on this approach will concentrate on verifying our model using more open datasets, including DEAP, DREAMER, and SEED. This system can accurately identify various emotions and be applied to sentiment analysis, medicine, and mood detection. Additionally, we want to reliably identify more varieties of emotions using EMOTIV EPOC X headbands to capture data and employ the proposed method. Additionally, we are interested in investigating multi-modal analysis for emotion classification.

## Figures and Tables

**Figure 1 sensors-22-08550-f001:**
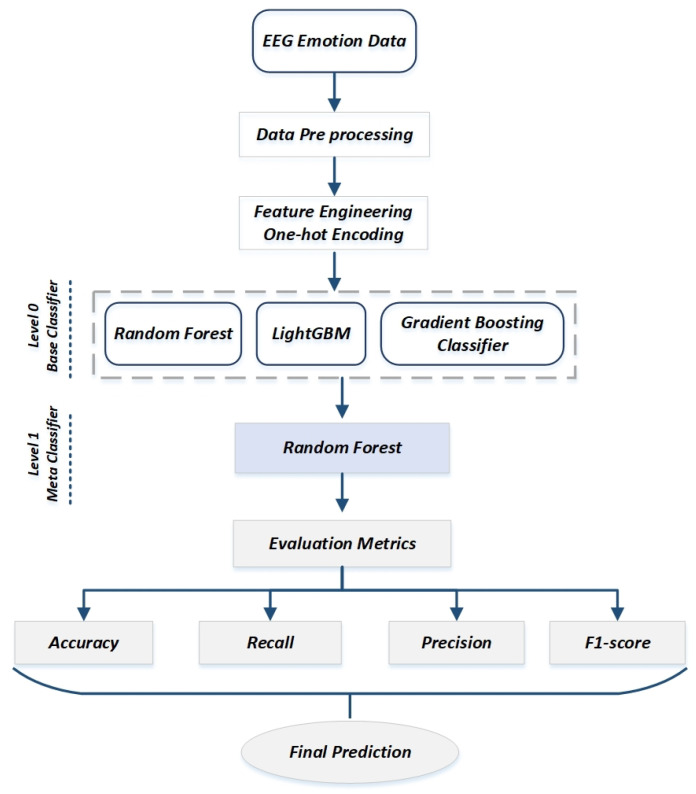
Base concept of proposed methodology.

**Figure 2 sensors-22-08550-f002:**
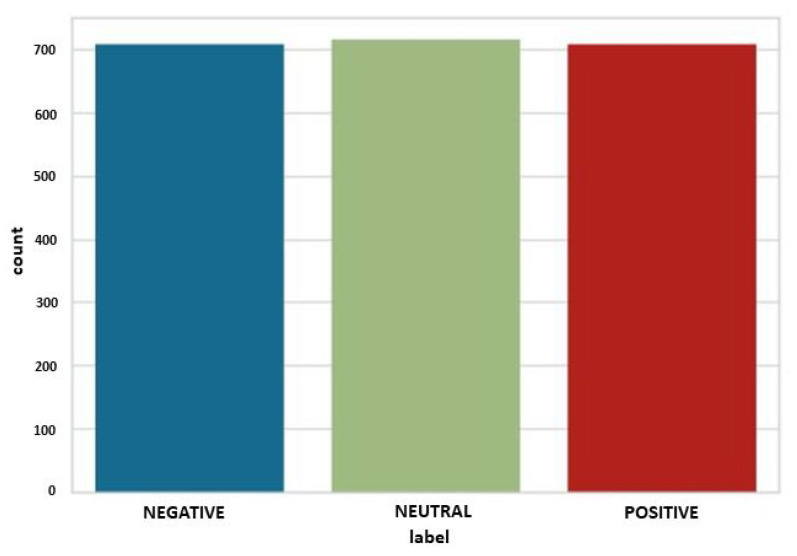
Visualization of three emotional data.

**Figure 3 sensors-22-08550-f003:**
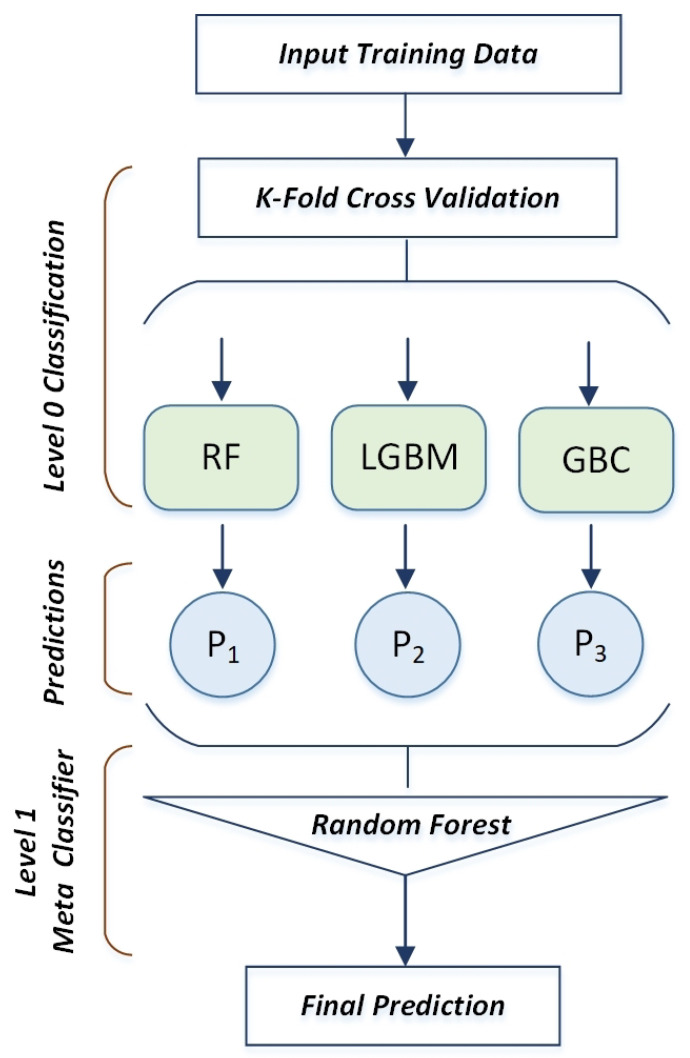
Stacking ensemble classification model architecture.

**Figure 4 sensors-22-08550-f004:**
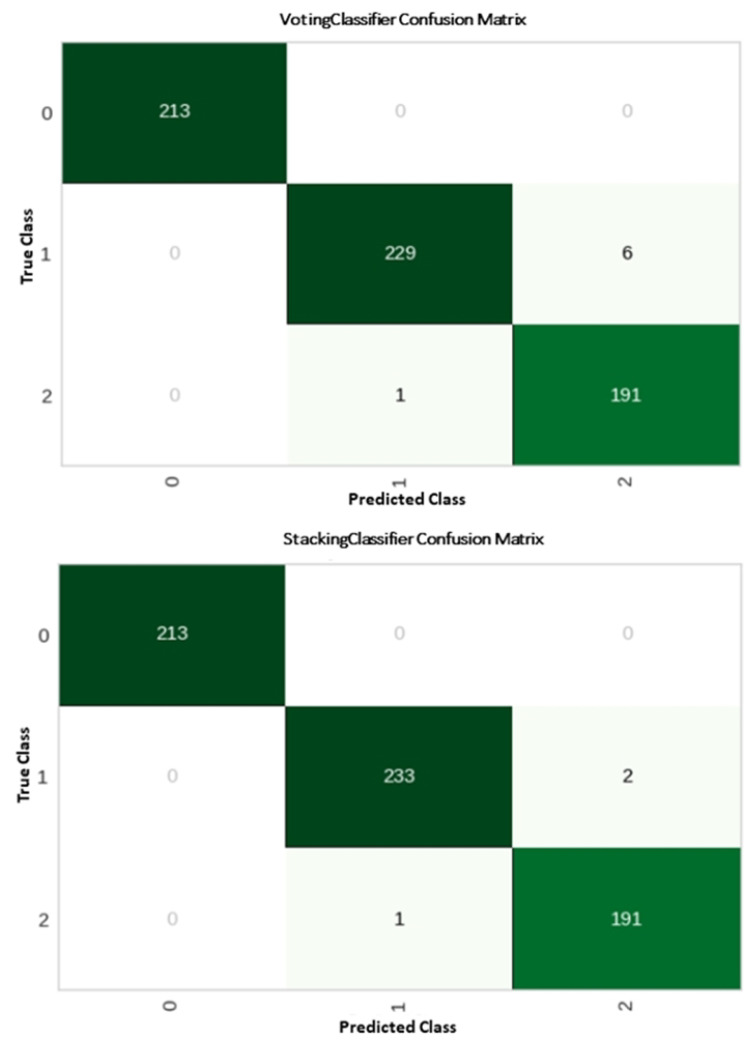
Confusion matrix for voting and stacking classifiers.

**Figure 5 sensors-22-08550-f005:**
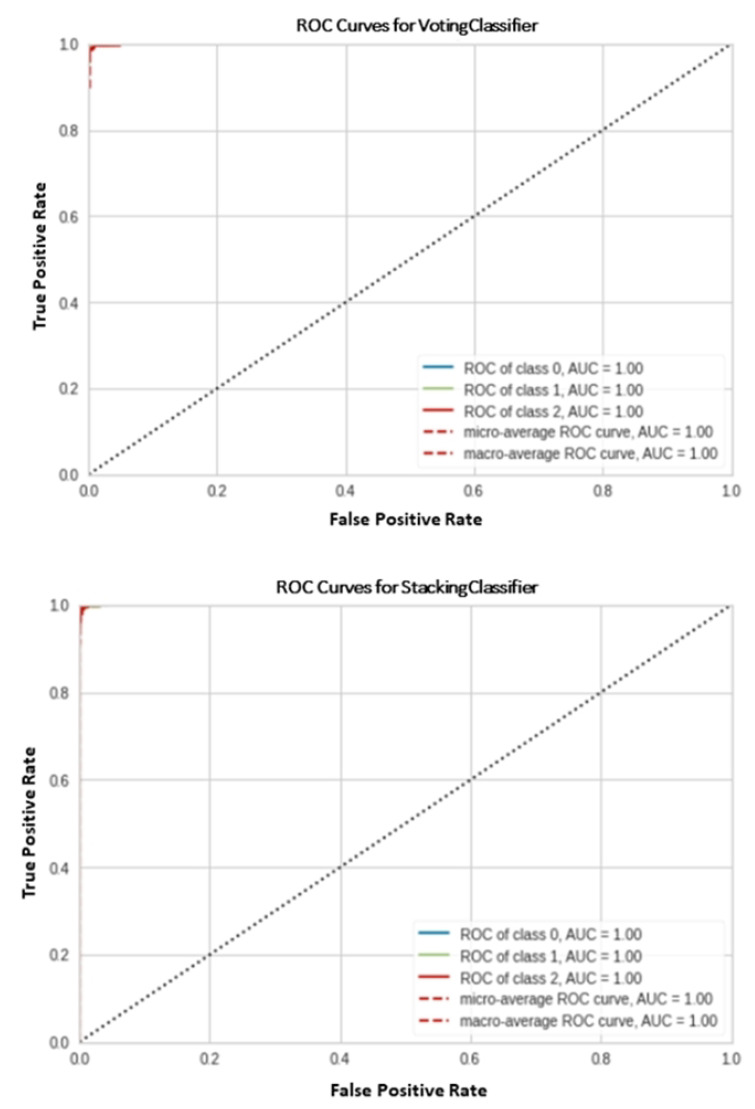
ROC curves for voting and stacking classifiers.

**Table 1 sensors-22-08550-t001:** The details used for EEG brainwave data collection.

Sl No.	Film Name	Emotion Labels	Scene	Studio Name	Year
1	Marley and Me	Negative	Death Scene	Twentieth Century Fox	2008
2	Up	Negative	Opening Death Scene	Walt Disney Pictures	2009
3	My Girl	Negative	Funeral Scene	Imagine Entertainment	1991
4	La La Land	Positive	Opening musical number	Summit Entertainment	2016
5	Slow Life	Positive	Nature timelapse	BioQuest Studios	2014
6	Funny Dogs	Positive	Funny dog clips	MashupZone	2015

**Table 2 sensors-22-08550-t002:** Final parameter settings for models.

Sl. No.	Parameter	Random Forest	LightGBM	GBC
1	learning_rate	-	0.2	0.3
2	bagging_fraction	-	0.6	-
3	bagging_freq	-	3	-
4	n_estimators	100	100	270
5	feature_fraction	-	0.9	-
6	num_leaves	-	90	-
7	min_samples_leaf	1	-	2
8	min_samples_split	2	-	7
9	max_features	auto	-	sqrt

**Table 3 sensors-22-08550-t003:** Comparison of evaluation matrices in terms of different classifiers.

Sl No.	Model	Accuracy	Recall	Precision	F-Measure
1	Gradient Boosting Classifier	0.9873	0.9870	0.9874	0.9873
2	Light Gradient Boosting Machine	0.9873	0.9870	0.9874	0.9873
3	Random Forest Classifier	0.9859	0.9857	0.9862	0.9859
4	Extra Trees Classifier	0.9765	0.9759	0.9773	0.9765
5	Decision Tree Classifier	0.9631	0.9626	0.9637	0.9631
6	Ada Boost Classifier	0.8297	0.8309	0.8604	0.8217
7	K Neighbors Classifier	0.7506	0.7447	0.7434	0.7418
8	Voting classifier	0.9891	0.9898	0.9893	0.9891
9	Proposed Stacking model	0.9955	0.9954	0.9953	0.9953

**Table 4 sensors-22-08550-t004:** Accuracy comparison with other models.

Methods	Accuracy
Random Tree	79.21
Deep Belief Network	88.66
Multi-layer Perceptron classifier	84.95
Recurrent Neural Network	87.65
Proposed method	99.55

## Data Availability

Not applicable.

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
