# Peer review of "EEG-Based Emotion Classification Using Stacking Ensemble Approach"

_sensors, 2022, doi:10.3390/s22218550_

Round 1

Reviewer 1 Report

Table.3
Please add bold attribute to row no.9
Please add references to particular models if it is possible

l.244
table <-Table

Figure.5
recommended square proportions of figures

Eq.4
/cdot operator instead * required

l.350
electrocorticography <- ElectroCorticoGraphy

Fig.2, 4, 5
Low quality, please use vector graphics

Reviewer 2 Report

This paper focuses on EEG-Based Emotion Classification by proposing a new approach based on stacking strategy. Experimental comparison shows superior performance of the proposed approach. Followings are my concerns:

1. There are numerous approaches for EEG Emotion Classification. The motivation of proposing new approach should be demonstrated more clearly.

2. Pseudocode should be included to show the proposed approach clearly.

3. More descriptions should be added to the proposed approach.

4. To show the importance of the EEG-based emotion classification, some related works and techniques can be added to enrich the manuscript. For example: A novel sample and feature dependent ensemble approach for Parkinson’s disease detection; EEG-based Emotion Recognition Using Hybrid CNN-LSTM Classification; Semisupervised Multiple Choice Learning for Ensemble Classification

5. The computation complexity of the proposed method should be clearly described.

6. Please correct the misprints in the manuscript. For example, it should be "Table 1".

7.  More in-depth analysis should be added in the experimental part.

Reviewer 3 Report

The paper needs to explain methodology part in detailed manner. For example, the feature extraction part needs more explanation. The Introduction part should include emotion recognition based on other physiological signals like ECG and should state the motivation for studying EEG signals for emotion identification. The paper should compare the method with other existing methods for emotion detection for example, multiwavelet transform based method, multivariate Fourier Bessel series expansion based empirical wavelet transform, and higher order statistics, etc.

What about computational complexity of the method? Comment on it.
